# A Scale Conversion Model Based on Deep Learning of UAV Images

Xingchen Qiu [1,2], Hailiang Gao [1,*], Yixue Wang [1,2], Wei Zhang [1], Xinda Shi [1], Fengjun Lv [3], Yanqiu Yu [3], Zhuoran Luan [3], Qianqian Wang [1,2] and Xiaofei Zhao [4]

1    National Engineering Laboratory for Satellite Remote Sensing Applications, Aerospace Information Research Institute, Chinese Academy of Sciences, Beijing 100101, China; qiuxingchen21@mails.ucas.ac.cn (X.Q.); wangyixue20@mails.ucas.ac.cn (Y.W.); zhangwei01@aircas.ac.cn (W.Z.); shixd@aircas.ac.cn (X.S.); wangqianqian22@mails.ucas.ac.cn (Q.W.)
2    University of Chinese Academy of Sciences, Beijing 100049, China
3    College of Earth Sciences, Hebei Geo University, Shijiazhuang 050030, China; lvfj1973@hgu.edu.cn (F.L.); yuyanqiu@hgu.edu.cn (Y.Y.); luanzhuoran@hgu.edu.cn (Z.L.)
4    College of Geo-Exploration Science and Technology, Jilin University, Changchun 130026, China; zhaoxf2319@mails.jlu.edu.cn
*    Correspondence: gaohl200439@aircas.ac.cn

**Abstract:** As a critical component of many remote sensing satellites and model validation, pixel-scale surface quantitative parameters are often affected by scale effects in the acquisition process, resulting in deviations in the accuracy of image scale parameters. Consequently, various successive scale conversion methods have been proposed to correct the errors caused by scale effects. In this study, we propose ResTransformer, a deep learning model for scale conversion of surface reflectance using UAV images, which fully extracts and fuses the features of UAV images in the sample area and sample points and establishes a high-dimensional nonlinear spatial correlation between sample points and sample area in the target sample area, so that the scale conversion of surface reflectance at the pixel-scale can be completed quickly and accurately. We collected and created a dataset of 500k samples to verify the accuracy and robustness of the model with other traditional scale conversion methods. The results show that the ResTransformer deep learning model works best, providing average MRE, average MRSE, and correlation coefficient R values of 0.6440%, 0.7460, and 0.99911, respectively, and the baseline improvements compared with the Simple Average method are 92.48%, 92.45%, and 16.59%, respectively. The ResTransformer model also shows the highest robustness and universality and can adapt to surface pixel-scale conversion scenarios with different sizes, heterogeneous sample areas, and arbitrary sampling methods. This method provides a promising, highly accurate, and robust method for converting pixel-scale surface reflectance scale.

**Keywords:** pixel scale; surface reflectance; scale conversion; deep learning; UAV

## 1. Introduction

Scale conversion is a critical link in many remote sensing physical modelling, remote sensing product applications and quantitative description of surface parameters at the pixel scale [1–4]. The primary problem in remote sensing scale conversion is effectively converting remote sensing data and information from one scale to another and simultaneously giving the evaluation index and uncertainty of the scale conversion results [2,5,6]. The pixel-scale surface reflectance is necessary for satellite verification and inversion of other quantitative remote sensing parameters [7]. In obtaining the surface reflectance at the pixel scale, the surface reflectance of the sampling points collected on the ground needs to be scaled up to the pixel scale via the point-to-surface scale conversion method [8–10].

The main methods of upscaling remote sensing parameters in the past decades include Simple Average, empirical regression, geostatistical, and Bayesian methods. Among them,

the Simple Average method calculates the arithmetic mean of multiple sample points or the weighted mean of each sample point according to the components within the sample area as the actual image element scale value, which is a simple method [11–13]. However, it only applies to areas with low heterogeneity of surface parameters. Otherwise, the results will bear significant uncertainty [14]. The empirical regression method establishes the interrelationship between the measured values of each sample point, the related metadata and the actual values of the image element scale through the historical observation samples. A series of regression models are then used such as the least squares method to fit [15–17], which has improved accuracy compared with Simple Average, but is overly dependent on historical observation data and has poor portability [18]. The geostatistical method aims to combine the theory of regional variation with the variation function as a tool to reconstruct the measured values of internal sampling points to the metric scale [19,20]. Moreover, the geostatistical method has no interference from other metadata, and its accuracy is only related to the distribution of the number of sampling points; however, this also leads to an overdependence of its result accuracy on the sampling method [21]. The Bayesian method adopts the Bayesian maximum entropy method [22], incorporates various prior information, and has no requirement for the distribution of prior knowledge [23]. However, this method is more complex, and the probabilistic transformation of various prior knowledge is still in the research stage.

Due to subsurface heterogeneity during the field measurement of quantitative remotely sensed surface parameters, the quantitative inversion models of the corresponding parameters and the accuracy of the results produce variations of different sizes. While validating the lake surface albedo model, Du et al. found that spatial heterogeneity led to lower precision of the point-scale reflectance due to the lack of scale conversion, which significantly impacted the accuracy of the validation results of the model [24]. The current research on scale conversion for many remotely sensed surface parameters mainly focuses on surface temperature, LAI, BRDF, total primary productivity and landscape phenology parameters. Liang et al. used high-resolution remote sensing images to upscale ground point-scale reflectance measurements to low-resolution images with reasonable accuracy [25]. Yue et al. used high-resolution remote sensing images to upscale ground point-scale albedo measurements to the 500 m image scale [26], and Sanchez-Zapero et al. combined the spatial heterogeneity of the subsurface. They employed geostatistical methods that combined the subsurface's spatial heterogeneity and scaled the observation tower's albedo measurements to the image element scale by the geostatistical method and the Simple Average method [27].

Due to the influence of surface heterogeneity, sampling method and randomness of sampling in the image element range, the scale conversion of the traditional point surface conversion method yields low accuracy of the pixel-scale surface reflectance, and the uncertainty of the scale conversion results fluctuates wildly [28]. For example, the traditional Simple Average method provides high accuracy of the image element-scale surface reflectance for sites with low surface heterogeneity, while the uncertainty of the scale conversion results is more significant for areas with high heterogeneity. Regarding the empirical regression method, due to the convenience of ground truth surface reflectance measurements, and based on the principle of incorporating as many types of features and multiple sample areas as possible [29], the sampling locations are often irregular and random, making it challenging to obtain historical observation samples [30]. The surface reflectance will change even in the same area, so the empirical model obtained in the previous period does not apply to scale conversion of the current environment. The geostatistical and general interpolation method requires a high number and distribution of sample points. It often performs better in areas with a high density of sample points. However, because the accessibility of different feature types in the same area differs, the number of sample points may be small and the distribution of sample points may be uneven [31]. Hence, the accuracy of the surface reflectance results obtained by the geostatistical method varies significantly in different types of sample areas, and the method's robustness is low. In

summary, the traditional point-to-surface conversion method is unsuitable for pixel-scale surface reflectance acquisition.

Therefore, this paper proposes an adaptive and highly robust ResTransformer surface-reflectance scale-conversion deep learning model that combines UAV images. The model extracts, fuses, and adaptively learns the features within the UAV image of the target sample area and the UAV image of the sampling points within the sample area through the ResTransformer deep learning model. It then establishes a high-dimensional nonlinear spatial correlation between the sampling points within the target sample area and the sample area, and maps the measured surface reflectance of each sampling point within the sample area to the surface reflectance of the sample area through the ResTransformer model. The surface reflectance of each sampling point within the sample area is mapped to the surface reflectance of the sample area by the ResTransformer model. It is possible to scale the surface reflectance of arbitrary size, heterogeneous sample area, arbitrary sampling location and the number of scenes with high-precision pixel scale.

The main objective of this study is to develop a deep learning model with UAV images to achieve high-accuracy pixel-scale surface-reflectance scale conversion for arbitrary size, heterogeneous sample area, arbitrary sampling location and the number of scenes. This method provides a relatively easy-to-use method to scale the surface reflectance, and the average MRE, average MRSE, and correlation coefficient R with the actual values on the large-scale dataset being 0.6440%, 0.7460 and 0.99911, respectively, 92.45% and 16.59%, respectively, compared to the baseline method. It is also found that the scale conversion accuracy of the deep learning model remains high for arbitrary size, heterogeneous sample area, arbitrary sampling location and the number of scenarios. Moreover, it demonstrates higher robustness and generalizability than the traditional methods. Section 2 of this paper introduces the normalization method for the reflectance scale conversion problem and the data source and construction of the large-scale reflectance scale conversion dataset. Section 3 introduces the ResTransformer deep learning model and the evaluation method of the accuracy of the scale conversion results. Section 4 analyzes and evaluates the accuracy of various scale conversion models on the dataset, and we conclude the above work in Section 5.

## 2. Reduction Method and Dataset

### 2.1. Reduction Method

Reduction refers to converting a complex problem or expression into a more straight-forward form that can be solved or processed more efficiently [32]. It is commonly used in computer science for solving problems in computational theory, such as proving that a problem is NP-complete and for transforming a complex problem into a known simple problem to be solved [33,34]. The prerequisites for reductio ad absurdum are that problem A can be transformed into problem B via rules and algorithms, and that both problems have the same type of answer, and the reduction is reversible and valid [35,36].

Since the process of imaging the sample area observation by UAV is similar to the process of detecting the reflectance of the sample point by the spectrometer above the sample point [37–39], the scale conversion problem of surface reflectance is considered to be reduced to the scale conversion problem of the grayscale value of the sampling point of the UAV image, as shown in Figure 1. The surface reflectance at the image element scale for any sample area N is the average of the reflectance of each continuous element within it [40]. However, measuring the reflectance of each continuous element in the field measurement is impossible, and the efficiency could be better [41]. Therefore, only the reflectance $Ref(n_k)$ of k sampling points inside the sample area N is usually measured, and the image-scale surface reflectance of the sample area is obtained by establishing a functional relationship $F$ between the reflectance of the sampling points and the reflectance $Ref(\text{N})$ of the sample area N, as shown in Equation (1).

$$Ref(\text{N}) = F(Ref(n_1), Ref(n_2), \dots, Ref(n_k)) \tag{1}$$

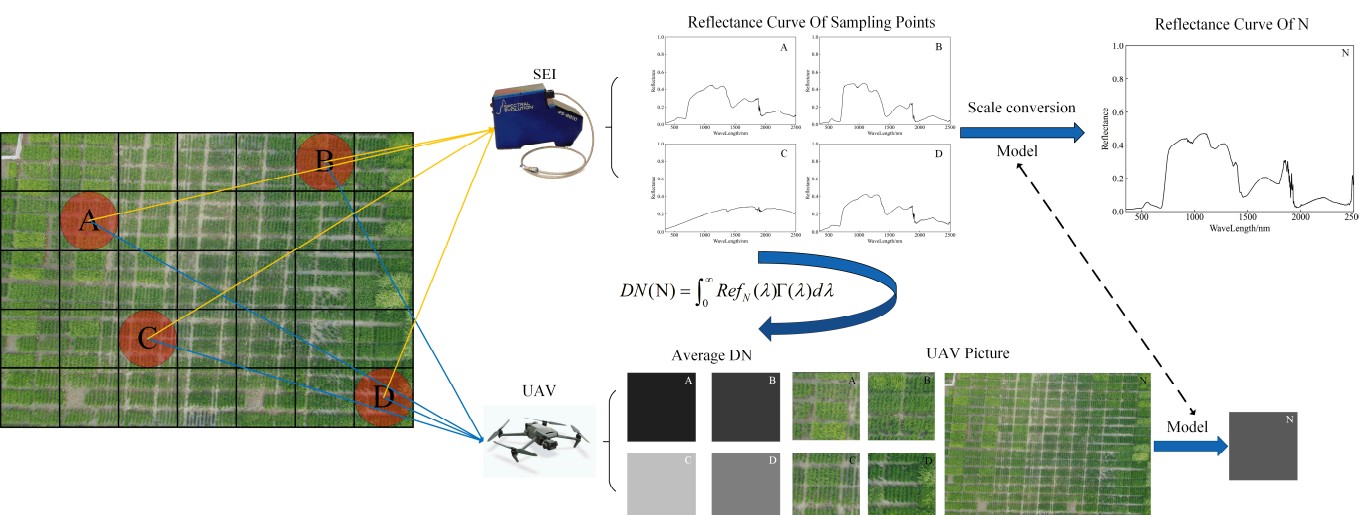

**Figure 1.** Schematic diagram of surface-reflectance scale conversion reduction. A,B,C,D are the numbers of sampling points respectively.

The panchromatic images obtained from UAV capture can simultaneously measure the grayscale values of each consecutive element within the sample area N. Therefore, the complex nonlinear correlation between all the sampling points in the sample area N and the sample area N is established by using the UAV images in the sample area, all the UAV images in the sampling points, and the average grey value $DN(\mathrm{N})$ of the UAV images in the sample area and the average grey value $DN(n_k)$ of the UAV images in the sampling points are calculated to optimize and verify $F$, as shown in Equation (2).

$$DN(\mathrm{N}) = F(DN(n_1), DN(n_2), \ldots, DN(n_k)) \tag{2}$$

Moreover, the result of the convolution of the surface reflectance of the sample area and the spectral response function of the UAV imaging spectrometer is related to the grayscale value of the UAV image so that Equation (3) exists, allowing arbitrary surface reflectance to be converted to the grayscale value of the UAV image in polynomial time, where $\Gamma(\lambda)$ is the spectral response function at the corresponding wavelength.

$$DN(\mathrm{N}) = \int_0^\infty Ref_N(\lambda)\Gamma(\lambda)d\lambda \tag{3}$$

In summary, the input of the pixel-scale surface reflectance scale conversion is converted into the input of the scale conversion of the UAV image of the sampling point using Equation (3). The features of the sample area and the UAV image of the sampling point are extracted by the deep learning model to obtain the feature representation of the corresponding space. The similarity weights of the features of the sample area and the sampling point are then obtained by feature fusion. After that, the average $DN$ value of the UAV images of the sampling points is multiplied by the corresponding feature weights, and the corresponding corrected sampling results are obtained through the fully connected layer calculation. The nonlinear correlation model between the UAV images of the sampling area and the UAV images within the sampling points is then established by mining the complex nonlinear spatial relationship between the effective sampling points and the sample area, solving the problem of scale conversion of the grayscale values of the sampling points of the UAV images, and the correlation model is applied to the reflectance of the sampled points to complete the normalization of the reflectance scale conversion problem.

*2.2. UAV Image Data*

A DJI (DJI Technology Co., Ltd., Shenzhen, China) Mavic 2 UAV took the UAV images used in this paper with a diagonal field of view of 84° and a single shot of

3000 px × 4000 px in actual size. Our team travelled to Xiong'an, Hebei, Shihezi, Xinjiang, Yuanmou, Yunnan, and Sanya, Hainan, between 2022 and 2023 to capture feature types specific to the destinations. Figure 2 shows the image locations and feature types. Table 1 shows the number of UAV images of each feature type and the observation area corresponding to the UAV images.

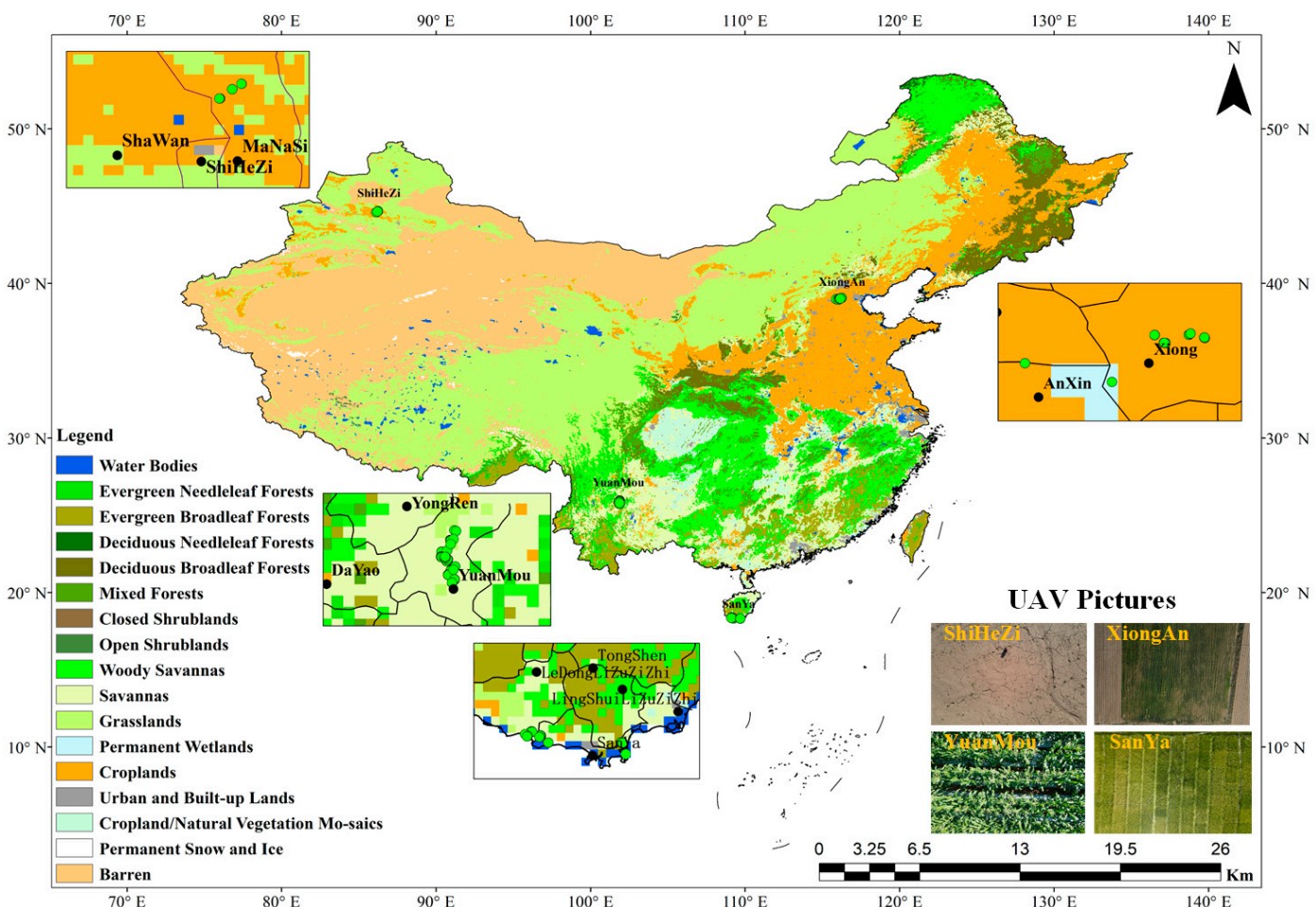

**Figure 2.** UAV image acquisition location and feature type.

**Table 1.** Number of UAV images of different feature types and corresponding observation areas.

| Feature Type | Num | Area km$^2$ | Feature Type | Num | Area km$^2$ | Feature Type | Num | Area km$^2$ |
|---|---|---|---|---|---|---|---|---|
| Chili | 56 | $1.83 \times 10^0$ | Corn | 211 | $6.09 \times 10^0$ | Sweet potato | 36 | $2.16 \times 10^0$ |
| Rice | 143 | $1.52 \times 10^1$ | Eggplant | 6 | $3.08 \times 10^1$ | Orange trees | 42 | $1.10 \times 10^0$ |
| Wheat | 52 | $1.56 \times 10^1$ | Cauliflower | 48 | $1.26 \times 10^0$ | Water bodies | 73 | $3.69 \times 10^0$ |
| Grassland | 136 | $4.92 \times 10^0$ | Peanut | 11 | $2.00 \times 10^1$ | Chrysanthemum | 20 | $8.02 \times 10^1$ |
| Cotton | 8 | $3.01 \times 10^1$ | Soybean | 15 | $3.21 \times 10^1$ | green cabbage | 12 | $2.35 \times 10^0$ |
| Zucchini | 74 | $3.33 \times 10^0$ | Sandy | 56 | $8.36 \times 10^0$ | Lover's Grass | 88 | $1.96 \times 10^0$ |
| Concrete | 65 | $7.38 \times 10^0$ | Tomatoes | 222 | $6.14 \times 10^0$ | Fluffy grass | 185 | $4.87 \times 10^0$ |
| Greenhouse | 86 | $4.94 \times 10^0$ | Bare soil | 482 | $3.39 \times 10^1$ | Golden peach | 60 | $3.63 \times 10^0$ |
| Okra | 5 | $3.40 \times 10^1$ | Date palm | 2 | $3.53 \times 10^1$ | Purple kale | 111 | $2.89 \times 10^0$ |
| Pitch | 16 | $1.14 \times 10^1$ | Open space | 2 | $2.68 \times 10^1$ | Green onions | 132 | $5.98 \times 10^0$ |
| Beans | 14 | $2.82 \times 10^0$ | Gravel | 2 | $3.15 \times 10^1$ | | | |
| Weeds | 12 | $7.87 \times 10^2$ | Straw | 47 | $2.77 \times 10^0$ | | | |

*2.3. Dataset Construction*

A large number of a priori samples are needed to extract the features and correlation between the corresponding sample areas and the sampled UAV images by deep learning models and to solve the scale conversion problem of the grayscale values of the sampled UAV images, for which the construction of a large-scale dataset is one of the critical steps [42,43]. In this paper, we build a large-scale scale-conversion dataset (SCD) containing multiple feature types, sample area sizes and sampling methods based on the UAV images acquired from several field experiments, and it includes 500k training samples. Figure 3 shows the overall process of SCD dataset construction.

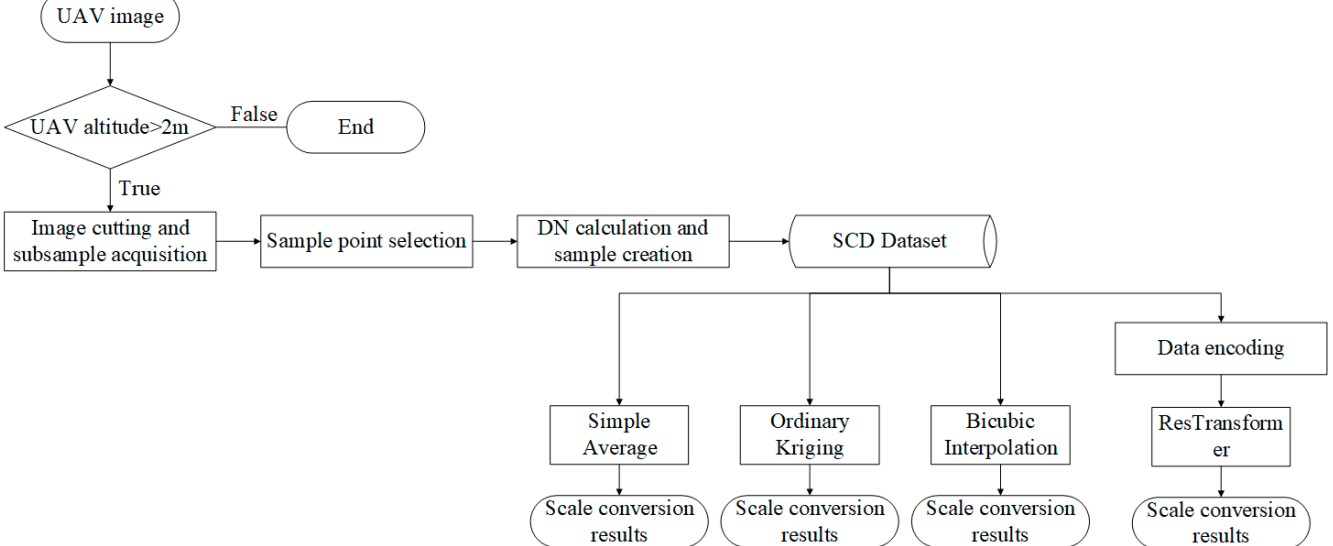

**Figure 3.** Overall process of SCD dataset construction.

The UAV images obtained from the field experiment acquisition first need to judge whether the image acquisition height is greater than 2 m to avoid the sample area being too small, leading to many overlapping simulated sampling points, which affects the accuracy of the data set. Since the surface reflectance acquisition and the pixel-scale surface reflectance acquisition process require the ground sample area to be square, and the UAV image is a rectangle of 3000 px × 4000 px, so for any UAV image, it needs to be cropped to a square of 3000 px in length and width. The SCD database is a large-scale pixel-scale reflectance scale conversion dataset containing multiple feature types, multiple sample area sizes, and multiple sampling methods. In addition, the deep learning model requires a large number of a priori samples, in order to increase the training samples, perform sample enhancement and achieve the purpose of simulating different high-spatial-resolution satellite pixel-scale size sample area scale conversion, by simulating UAV at different altitudes. The sample enhancement is achieved by simulating the UAV at different heights and imaging the sample area to obtain a variety of samples with different sample sizes. Figure 4 illustrates the overall process of sample enhancement. The spatial resolution of individual pixels of the original UAV image is calculated by Equation (4). The edge length of the square sample area under the simulated height h is calculated by Equation (5), in which $2h\tan(42^\circ)$ is the length of the diagonal of the corresponding enhanced sample because the cutting according to the long edge is prone to the problem of wide edge crossing, so the short edge cuts the enhanced sample as the square sample area. The short edge length is 0.6 times the diagonal length, and the corresponding sample area is calculated

by Equation (6). The number of pixel points in the edge length of the enhanced sample simulated UAV image $width_{px}$ is calculated by Equation (7).

$$resolution = \frac{2H\tan(42^\circ) \times \frac{3}{5}}{3000} = \frac{H\tan(42^\circ)}{2500} \tag{4}$$

$$width = 2h\tan(42^\circ) \times \frac{3}{5} \tag{5}$$

$$area = width \times width = \frac{36}{25}h^2\tan^2(42^\circ) \tag{6}$$

$$width_{px} = \frac{width}{resolution} = \frac{\frac{6}{5}h\tan(42^\circ)}{\frac{H\tan(42^\circ)}{2500}} = \frac{3000h}{H} \tag{7}$$

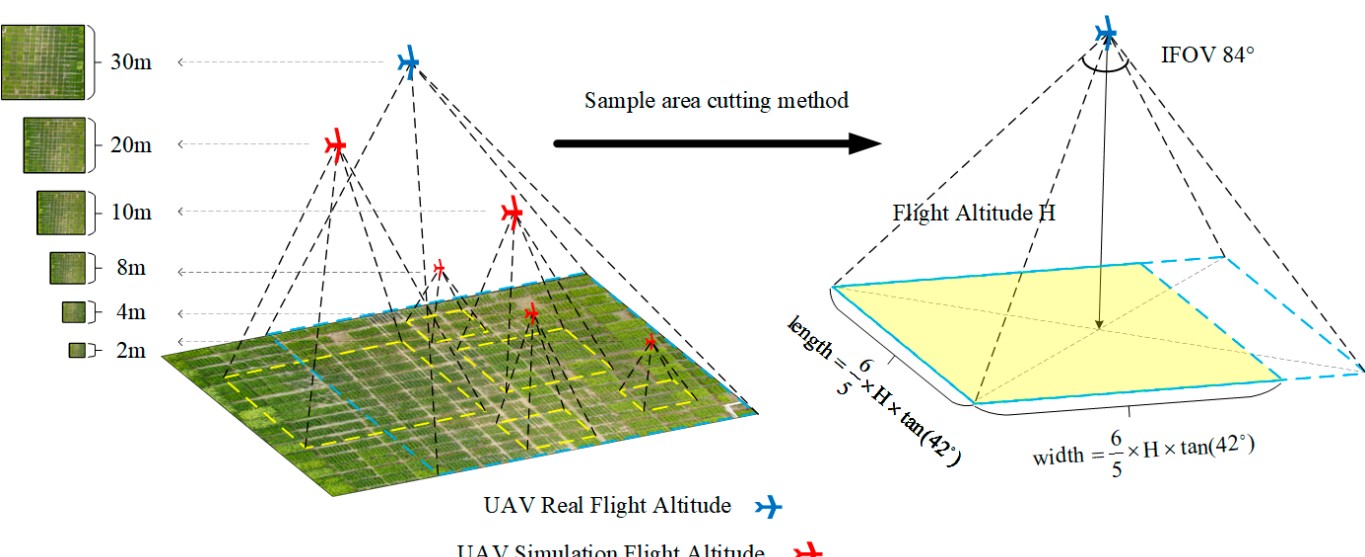

**Figure 4.** UAV image sample-enhancement scheme.

After obtaining the UAV images of the corresponding sample area through the above steps, it is necessary to simulate the field feature spectral acquisition process by selecting sampling points on the UAV images for simulated sampling. Sampling point selection includes standard and random sampling, and standard sampling adopts equal spacing sampling. The sampling methods include one-point sampling, two-point sampling, $2 \times 2$ four-point sampling, five-point sampling, $3 \times 3$ nine-point sampling and $4 \times 4$ sixteen-point sampling [44]. Random sampling uses randomly selected points, and the number of sampling points ranges from 1 to 16. Figure 5 shows the sampling locations of various sampling methods.

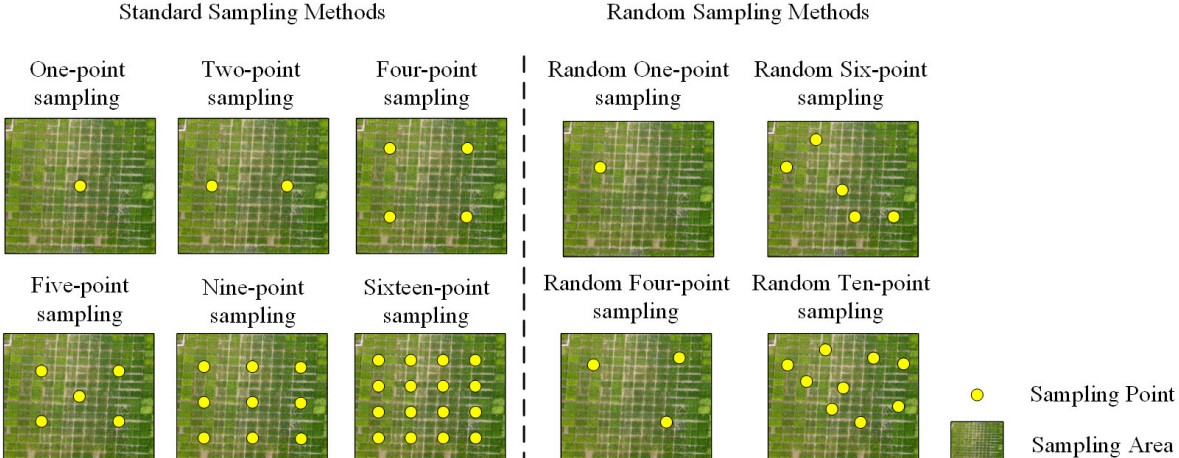

**Figure 5.** Schematic illustration of the location of sampling points for various sampling methods.

In order to make the samples in the dataset as close as possible to the actual field reflectance acquisition scene, we limit the acquisition height of each sampling point to 1 m~1.4 m by randomly selecting the sampling height $h_{point}$ as the sampling point. The side length $w_{point}$ of the corresponding sampling point UAV image is calculated by Equation (6). The corresponding sampling radius $r_{point}$ can be calculated by Equation (7), where $IFOV_{SEI}$ is the SEI spectrometer of IFOV. The IFOV in SCD was set to 25° because the field of view of the fiber optic probe of most of the mainstream spectrometers, such as SEI and ASD, is 25°.

$$w_{point} = \frac{5000 h_{point} \tan(IFOV_{SEI})}{h_{UAV} \tan(42°)} \tag{8}$$

$$r_{point} = \frac{h_{point} \tan(IFOV_{SEI})}{h_{UAV} \tan(42°)} \tag{9}$$

After obtaining the sample area UAV image and the sample location by simulated cropping and simulated sampling, the UAV image of the sample location is cropped. The average *DN* values of the sample point and the sample area UAV grayscale images are calculated, respectively, to complete the construction of the single SCD sample. Figure 6 shows the sample construction process. Since then, a single sample in the dataset has been created, which contains the UAV images of the sample area and the sample point and the average *DN* values of the UAV grayscale images of the sample area and the sample point.

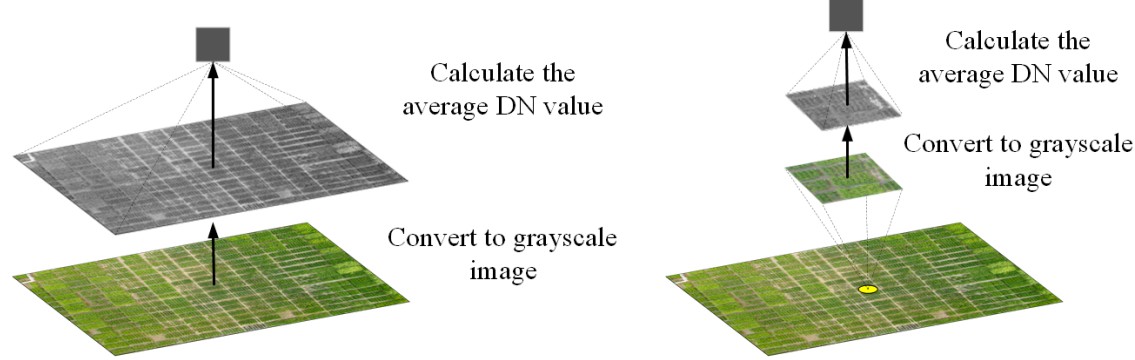

**Figure 6.** Construction process for a single sample of the SCD dataset.

Since the ResTransformer model is not an end-to-end model, it requires the same input data shape. However, in the process of surface reflectance measurement, due to various natural factors such as topography, geomorphology and subsurface state, the size of each sample area, the number, location and sampling height of sampling points are different, resulting in different resolutions of UAV images of sampling points within the sample area and the number of average *DN* values of sampling points. Hence, it is necessary to unify the samples in the SCD dataset by coding each. The samples in the SCD dataset need to be coded into the same structure.

Each sample in the SCD dataset includes the UAV images of the sample area, the sample points, and the *DN* mean of the UAV grayscale images. At the same time, for the UAV images of the sample points, since the number of sample points is usually less than 16 in the ground-based spectral acquisition process, we reserve 16 filled areas for the sample points and place the sample point images into the squares along the image channel dimension, each of which has a size of 56 px × 56 px. Similarly, the average grey value of the sampled UAV images is filled into a matrix of length 16 by cyclic mode-taking, and Figure 7 illustrates the sample normalization process.

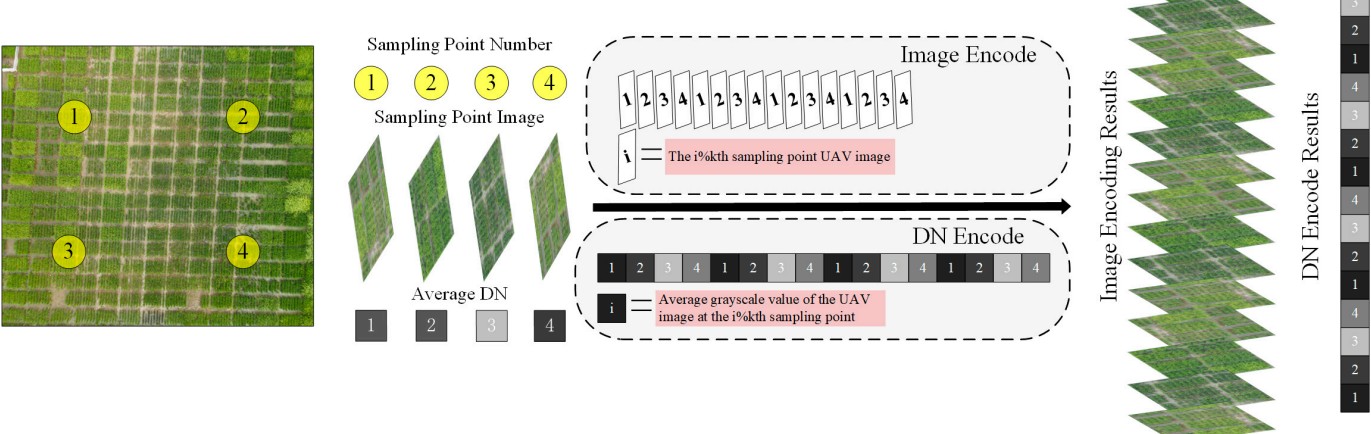

**Figure 7.** SCD dataset sample encoding method.

### 3. Methods and Evaluation Indicators

*3.1. Traditional Scale Conversion Methods*

The traditional scale conversion methods include Simple Average, Ordinary Kriging, and Cubic Spline Interpolation. Among them, the Simple Average method uses Equation (8) to calculate the mean value of *DN* values of each sampling point in the sample area as the scale conversion result of the sample area. Where *n* is the number of sample points in the SCD data set, $DN_{point\_i}$ is the average *DN* value of the UAV grayscale image at the *i*th sample point.

$$result_{simple\_average} = \frac{1}{n}\sum_{i=1}^{n} DN_{point\_i} \tag{10}$$

The Ordinary Kriging method first creates a variance function and a covariance function to estimate the spatial autocorrelation value. In this paper, a Gaussian function is used as the variance function. Equation (9) shows the mathematical model of the Ordinary Kriging method, where $Z^{\#}(x)$ is the unbiased optimal estimate of the *DN* value of the UAV grayscale image at location (x,y). $Z_i(a = 1, 2, \cdots n)$ is the mean value of *DN* of the UAV grayscale image at the sampling point. According to the two conditions of unbiased and optimality, the estimation equation of the weight coefficient $\lambda_i$ is shown in Equation (10). The solution of the equation shown in Equation (10) is brought into Equation (9) to obtain the estimated value of *DN* at the coordinates (x,y). Finally, the scale conversion result of

the sample is obtained by calculating the mean value of all *DN* values within the sample area [45–47].

$$Z^{\#}(x) = \sum_{i=1}^{N} \lambda_i Z_i \tag{11}$$

$$\begin{cases} \sum_{\beta=1}^{N} \lambda_\beta \gamma(x_\alpha, x_\beta) + \sum_{L=0}^{K} \mu_L f_L(x_\alpha) + \mu_0 = r(x_0, x) \\ \sum_{\alpha=0}^{N} \lambda_\alpha f_L(x_\alpha) = f_L(x) \\ \sum_{\beta=1}^{N} \lambda_\beta \gamma(x_\alpha, x_\beta) \end{cases} \tag{12}$$

The cubic Spline Interpolation method is a smooth surface interpolation method for estimating the function values between given data points. Similar to the one-dimensional case, it divides the surface between a given data point into multiple cubic function slices and ensures that the function and derivative values between adjacent slices are equal, thus ensuring the continuity and smoothness of the surface. Equation (11) shows the general form of Cubic Spline Interpolation, where $DN(x,y)$ is the *DN* value of the whole sample plane, $w_{i,j}$ is the weight, $N_i(x)N_j(y)$ is the basis function, and *m* and *n* are the total numbers of sampled points in the length and width direction of the sample area, respectively. The basis function is usually three-times the spline function; they have good smoothness and approximation properties. Equation (12) defines the spline function, where $h_i$ is the distance between two sampling points [48–50].

$$DN(x,y) = \sum_{i=1}^{m} \sum_{j=1}^{n} w_{i,j} N_i(x) N_j(y) \tag{13}$$

$$N_i(x) = \frac{1}{6h_i} \begin{cases} (x - x_{i-1})^3 & , x_{i-1} \le x < x_i \\ 3h_i^2(x - x_{i-1})^2 - 3h_i(x - x_{i-1})^3 + h_i^3 & , x_i \le x < x_{i+1} \\ 3h_i^2(x_{i+1} - x)^2 - 3h_i(x_{i+1} - x)^3 + h_i^3 & , x_{i+1} \le x < x_{i+2} \\ (x_{i+2} - x)^3 & , x_{i+2} \le x \end{cases} \tag{14}$$

*3.2. ResTransformer Deep Learning Model*

The ResTransformer model proposed in this paper is a deep learning framework which aims to scale the conversion of surface reflectance by extracting the features of the UAV images of the learning sampling points and the UAV images of the sample area and establishing the correlation between the average *DN* values of the UAV images of the sampling points and the average *DN* values of the UAV images of the sample area. The ResTransformer model consists of Resnet and Swim Transformer V2. ResNet is a deep residual network that introduces the residual block concept to add input and output data to learn residuals [51]. This residual learning approach can solve the gradient disappearance or gradient explosion problem encountered by neural networks during training and is widely used in computer vision tasks. Transformer is a deep learning model based on a self-attentive mechanism, initially proposed by Vaswani et al. in 2017 [52], whose core idea is encoding and decoding sequences through a multi-headed attention mechanism. The Transformer model first achieved excellent results in natural language processing [53,54]. With the Vision proposed by Dosovitskiy et al. in 2020 to achieve an image serialization Transformer model, Transformer has also been gradually applied to computer vision [55]. A successive series of Transformer-based computer vision models have been proposed, such as DETR and Swin Transformer [56].

The ResTransformer first extracts the features of the sample area and the sample point UAV images, obtains the spatial representation of the corresponding features and then uses feature fusion to obtain the similarity weights $\lambda$ of the sample area and the sample point features. The average *DN* value of the sample points multiplies the corresponding feature

weight $\lambda$ to obtain the corresponding corrected sampling results and establish the complex non-linear spatial relationship $f$ between the effective sample points and the sample area, after which the corresponding scale conversion results are obtained by Equation (13).

$$result_{\text{ResTransformer}} = f(\lambda_1 DN_1, \lambda_2 DN_2, \cdots \lambda_n DN_n) \tag{15}$$

ResTransformer combines the advantages of ResNet and Transformer, using ResNet-18 as the backbone network for feature extraction of sample area and sample point UAV images. We use a double-headed Resnet Block to extract features from the sample area and sample point UAV images. The sample area image size is $224 \times 224 \times 3$, so 12 Resnet Blocks are used to compress the feature size to $14 \times 14 \times 256$ after four times of downsampling, and the sample point UAV image size is compressed to $14 \times 14 \times 64$ by 4 Resnet Blocks after two times of downsampling, after which the sample area and sample point UAV image features are stitched according to the channel dimension.

The above output image features are passed through four Resnet Blocks and fused after one downsampling. The typical features are then extracted and computed using an attention mechanism by the Swin Transformer Block V2 module [57]. The input UAV image features in Swin Transformer Block V2 are split into multiple patches, each transformed independently. The different patches are connected by Cross-Stage Connection (CSC) and Local Window Swapping (LWSW). The information is exchanged and integrated between the patches through CSC and LWS. After fusing the output features of the encoder with the input of the linear layer, the complex non-linear spatial relationship between the sample area and the sampled points is obtained through a multi-layer full connection and a non-linear combination of activation functions.

The above output image features are passed through four Resnet Blocks and fused after one downsampling. After that, the typical features are extracted and computed by the Swin Transformer Block V2 module using the attention mechanism. The input UAV image features in Swin Transformer Block V2 are partitioned into multiple patches, and each patch is independently transformer computed. The different patches are connected by Cross-Stage Connection (CSC) and Local Window Swapping (LWSW). The information is exchanged and integrated between different patches through CSC and LWS. After fusing the output features of the encoder with the input of the linear layer, the complex nonlinear spatial relationship between the sample area and the sampling points is obtained by the nonlinear combination of the multilayer full connection and the activation function, and the scale transformation results are obtained by combining the above Equation (13). Figure 8 illustrates the overall architecture of ResTransformer.

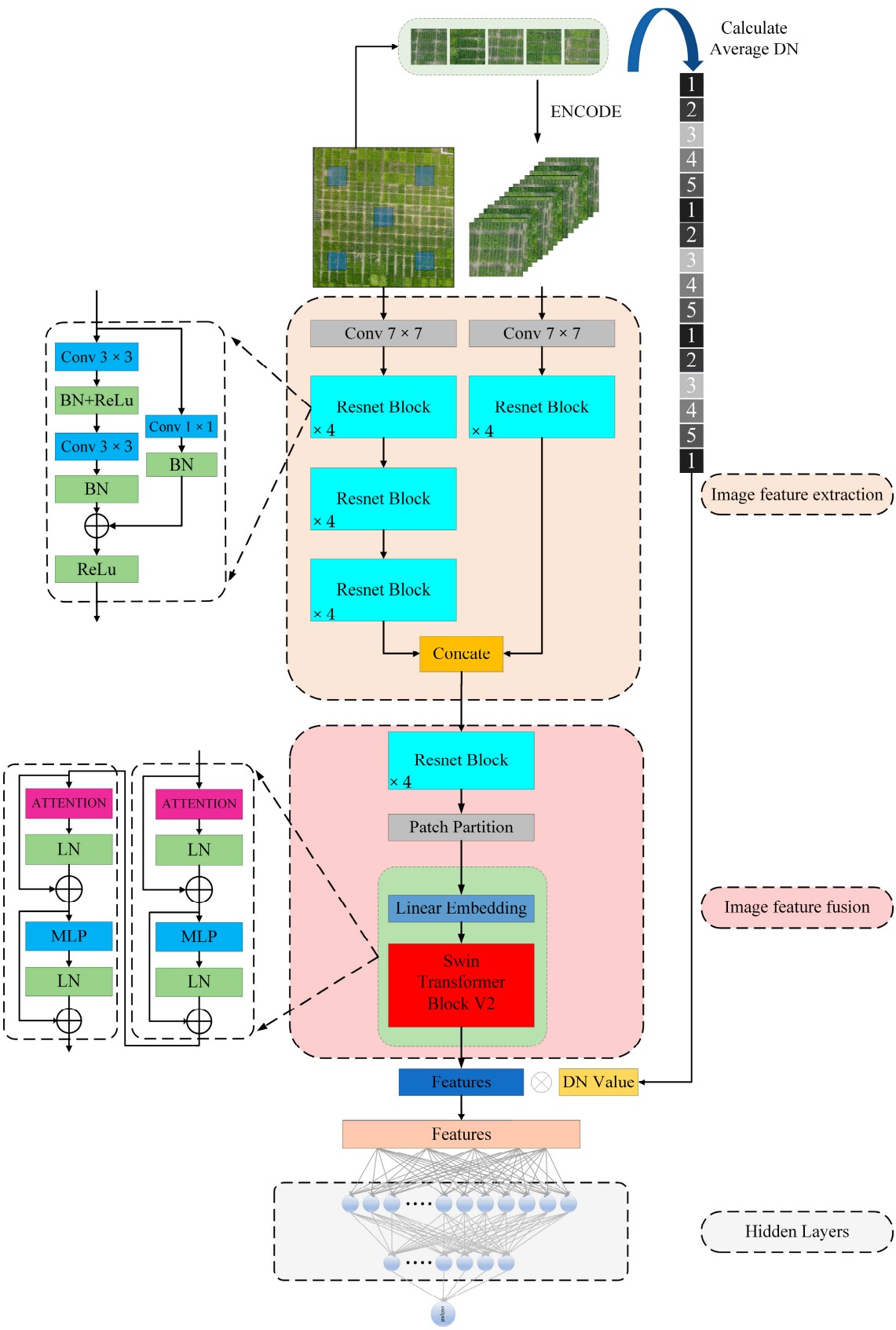

**Figure 8.** ResTransformer model structure.

To quantitatively describe the computational complexity of the ResTransformer model, we calculated the Floating-Point Operations (Flops) for each module of the ResTransformer model and the parameters of the corresponding modules to measure the computational complexity of the model. Table 2 shows the complexity of each component module. In the table, Head and Head small correspond to the input layer of the sample area UAV image and sample point UAV image, respectively; Resnet Blocks 1–3 are the sample area UAV image feature extraction layers; Resnet Block 2 small corresponds to the sample point UAV image feature extraction layer; Resnet Block 4 is the sample area, and sample point UAV image feature fusion. The Swin Transformer Block v2 is the function layer of feature fusion; Linear is the linear output layer.

**Table 2.** Complexity of each module of the ResTransformer model.

| Name | Input Shape | Output Shape | Flops | Flops Percentage | Params |
|---|---|---|---|---|---|
| Head | [3, 224, 224] | [64, 128, 128] | $1.20 \times 10^8$ | 6.008% | $9.54 \times 10^3$ |
| Head Small | [48, 56, 56] | [64, 28, 28] | $1.19 \times 10^8$ | 5.973% | $1.51 \times 10^5$ |
| Resnet Block 1 | [64, 128, 128] | [64, 56, 56] | $4.64 \times 10^8$ | 23.327% | $1.48 \times 10^5$ |
| Resnet Block 2 | [64, 56, 56] | [128, 28, 28] | $4.12 \times 10^8$ | 20.706% | $5.26 \times 10^5$ |
| Resnet Block 3 | [128, 28, 28] | [256, 14, 14] | $4.12 \times 10^8$ | 20.676% | $2.10 \times 10^6$ |
| Resnet Block 2 Small | [64, 28, 28] | [64, 14, 14] | $4.27 \times 10^8$ | 21.467% | $8.72 \times 10^6$ |
| Resnet Block 4 | [320, 14, 14] | [512, 7, 7] | $2.99 \times 10^7$ | 1.500% | $1.52 \times 10^5$ |
| Swin Transformer Block v2 | [49, 256] | [49, 256] | $6.81 \times 10^6$ | 0.342% | $1.51 \times 10^5$ |
| Linear | [49, 256] | [1] | $1.75 \times 10^4$ | 0.001% | $1.75 \times 10^4$ |
| TOTAL | - | - | $1.99 \times 10^9$ | 100% | $1.20 \times 10^7$ |

The results in the table show that the overall Flops of ResTransformer are $1.99 \times 10^9$, and the total number of parameters is $1.20 \times 10^7$, while the Flops of the traditional Resnet50 model are around $2.91 \times 10^9$. The Flops of the ResTransformer model are 31.69% lower compared to Resnet50. Our proposed ResTransformer model has higher computational efficiency and lower complexity.

### 3.3. Hyperparameter Setting

The traditional Resnet residual network is good at solving the gradient disappearance and gradient explosion problems in deep network training. However, as its network layers increase, there is a degradation problem of network accuracy. To solve the problem that it is challenging to optimize the deep network, we use the Batch Normalization layer by adding it to accelerate the model convergence and improve the accuracy and generalization ability of the model. Furthermore, for the parameters in Resnet, we use a normal distribution for initialization, which achieves good accuracy on ImageNet. The Transformer architecture in the computer vision neighborhood is prone to model overfitting and bias due to the lack of structural bias and many parameters. To solve the problem of Swin Transformer Block v2 overfitting during training, we only use a single Swin Transformer Block v2 to solve the overfitting problem of Swin Transformer Block v2, we only use a single Swin Transformer Block v2 to calculate the output features of Resnet, and at the same time, we add a Dropout module to Swin Transformer Block v2 and use LayerNorm to normalize the output layer features in order to improve the generalization ability and robustness of the model.

During the overall training of the ResTransformer model, we generated enough sample data to improve the accuracy and generalization ability of the model. We used a 7:3 ratio to divide the training and test sets. A batch size of 32 was used during the training process; compared to a large batch size, training with a small batch size tends to converge to flat minimization regions that tend to generalize better because they are more robust to changes between the training and test sets [58]; epoch sets to 100, and the early stop strategy is activated, so that when the accuracy of the model on the test set has still not improved after three epochs, the training is stopped early. The model is considered to have completed

optimization to reduce the risk of overfitting. The learning rate strategy is set to automatic decay learning rate strategy, the initial learning rate is 0.1, the focus round is set to 2, the decay factor is set to 0.8, and the cold start strategy is also configured. The variable learning rate can accelerate the convergence of the model; the optimizer adopts Adam optimizer, the weight decay of the optimizer is set to 0.0001, and the weight decay coefficient of L2 regularization is used to improve the convergence speed of the model. All input UAV images and *DN* values are divided by 255 to normalize their values to between [0, 1], and a fixed threshold is used to crop the gradient in the backpropagation process. The platform's hardware, on which the training model is carried out in this paper, contains an Intel Xeon Gold 6244 CPU, 128 G RAM, and an Nvidia GeForce RTX 3090 series 24 G graphics card.

*3.4. Evaluation Indicators*

In this paper, the Simple Average method is employed as the baseline of the dataset, and Cubic Spline Interpolation and Kriging interpolation (Gaussian kernel function) methods are used on the dataset to verify the accuracy of various scale conversion methods, respectively. The results are compared with the results of the ResTransformer model. The mean relative error (MRE) between the *DN* value of the sample area and the mean *DN* value of the sample area UAV images obtained by each scale transformation method is calculated by Equation (4), where $DN_{pred}$ is the *DN* value of the sample area UAV grayscale images obtained by the scale transformation method, $DN_{true}$ is the mean *DN* value of the sample area UAV grayscale images in the SCD dataset, and the root mean square error (RMSE) using Equation (5) is calculated to evaluate the accuracy of individual samples in the SCD dataset. The overall accuracy of each scale conversion method is evaluated by calculating the correlation coefficient $\rho$ by Equation (6), where $Cov(DN_{pred}, DN_{true})$ is the covariance between the scale conversion result and the actual value in the data set, and $\sigma DN_{pred}$ and $\sigma DN_{true}$ are the variances between the scale conversion result and the actual value, respectively. In summary, we verify the accuracy of various scale conversion methods using the above three methods.

$$MRE = \sum_{i=0}^{N} \frac{\left| DN_{pred}(i) - DN_{true}(i) \right|}{DN_{true}(i)} * 100\% \qquad (16)$$

$$RMSE = \sqrt{\frac{1}{N} \sum_{i=1}^{N} (DN_{pred}(i) - DN_{true}(i))^2} \qquad (17)$$

$$\rho = \frac{Cov(DN_{pred}, DN_{true})}{\sigma DN_{pred} \sigma DN_{true}} \qquad (18)$$

## 4. Results and Discussion

*4.1. Accuracy Verification of Various Scale Conversion Methods*

In this paper, the simple arithmetic averaging method, Cubic Spline Interpolation method, Ordinary Kriging square and ResTransformer deep learning model are used to validate the validation set of the SCD dataset, respectively. The results of the simple arithmetic averaging method are used as the baseline. Figure 9 shows the validation results, in which the scatter plot is plotted with the scale transformation results of each scale transformation model as the horizontal axis, the actual values of the SCD dataset as the vertical axis and the distribution are linearly fitted using the least squares method. The proper function of the conversion results for each scale shows that the slope of the fit function of the scale conversion result of the ResTransformer model is close to 1, and its scale conversion accuracy is the best; the slope of the fit function of the Ordinary Kriging method and that of the Simple Average method are both 0.72. The difference is less than 0.6%. Therefore, the results of the Ordinary Kriging method and the Simple Average method achieve similar accuracy on most of the data sets. The results of the Cubic Spline

Interpolation method are more different from the actual values, and the distribution of the results is very discrete, so the method is less effective.

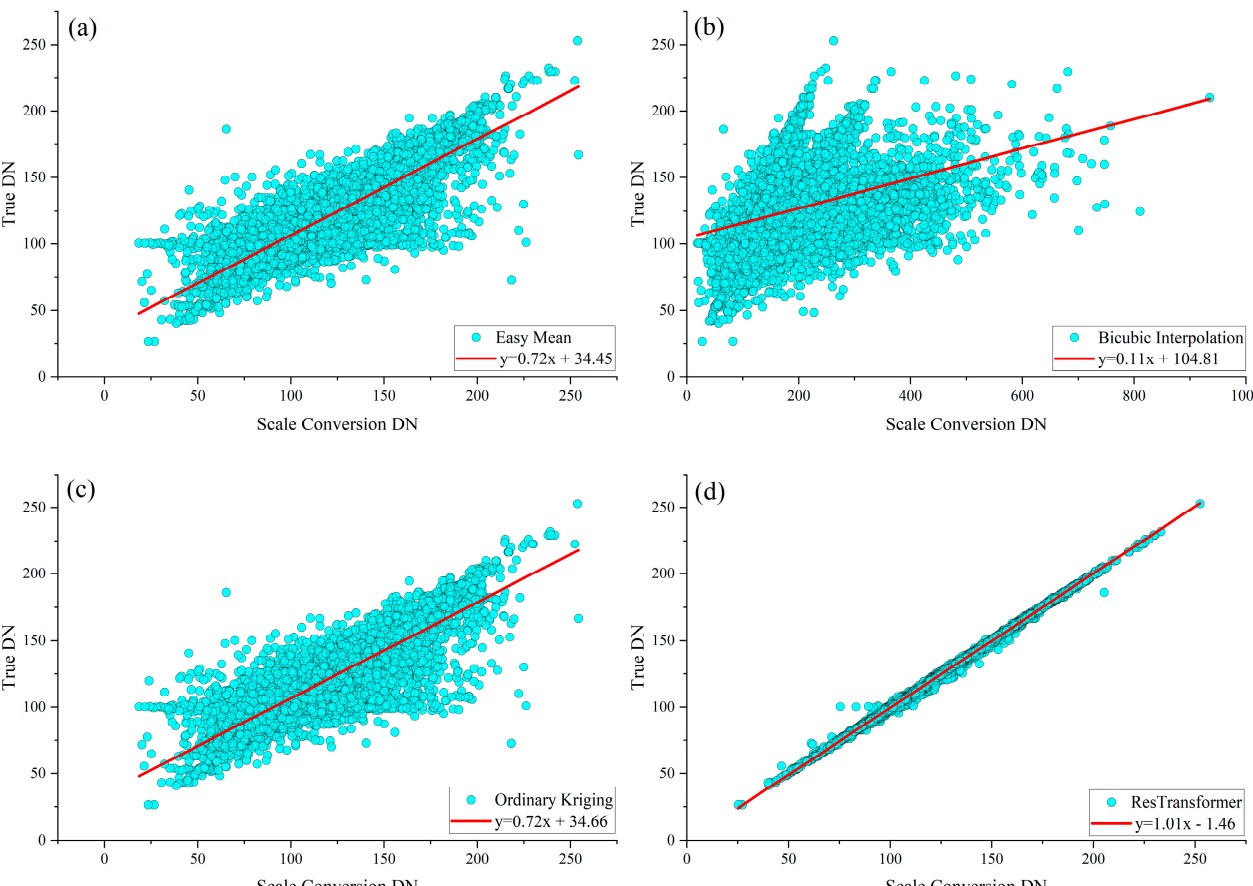

**Figure 9.** Scale conversion results for multiple scale conversion methods (**a**) Simple Average method, (**b**) Cubic Spline Interpolation method, (**c**) Ordinary Kriging method, (**d**) ResTransformer model.

The MRE and RMSE of the scale conversion results of various scale conversion methods and the actual values of the SCD dataset were calculated separately to measure the accuracy of various scale conversion methods. Figure 10 shows the results, and the horizontal and vertical coordinates in the figure are the MRE and RMSE of the scale conversion results of various methods, respectively. The results show that the conversion accuracy of the Ordinary Kriging method is similar to that of the Simple Average method in most scenarios. The MRE and RMSE of most samples are concentrated in the ranges of 0.3–20% and 0.4–20%, respectively. The results of the Cubic Spline Interpolation method are much less accurate than the baseline of the Simple Average method in both MRE and RMSE metrics, and the distributions of MRE and RMSE metrics are more discrete. The ResTransformer deep learning model outperforms the traditional surface reflectance scale conversion method in MRE and RMSE, where the MRE results are less than 4%, and most of the RMSE results are less than 4. Therefore, the scale conversion accuracy of the ResTransformer model is better than the baseline of the Simple Average method.

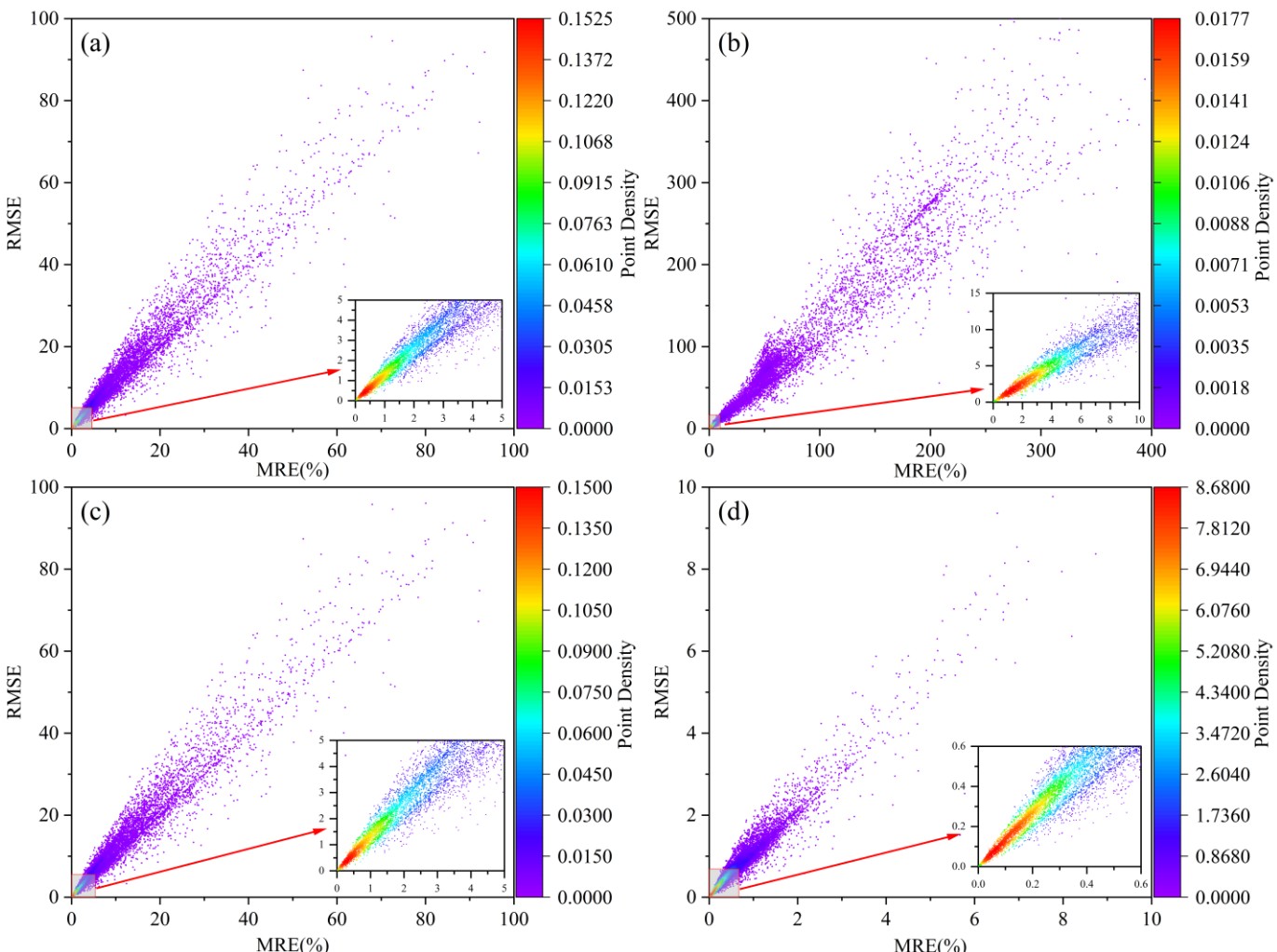

**Figure 10.** Accuracy of scale conversion results of multiple scale conversion methods (**a**) Simple Average method, (**b**) Cubic Spline Interpolation method, (**c**) Ordinary Kriging method, (**d**) ResTransformer model.

Since there are a few samples with abnormally high MRE and RMSE metrics in the results of various scale conversion methods, there are some shortcomings in evaluating the accuracy of the corresponding scale conversion methods by using the mean values of MRE and RMSE of the scale conversion results on the SCD dataset, and Figure 11 shows the distribution of the accuracy of different scale conversion results. In order to evaluate the accuracy of various scale conversion methods as a whole, the accuracy of each scale conversion method was evaluated by calculating the median, the overall mean, and the mean value within the range of non-abnormal values (Equation (17), where $Q_i$ is the i × 25th percentile of the data), and the correlation coefficient between the scale conversion results and the actual values in the SCD dataset, respectively, of the MRE and RMSE metrics of the results of various scale conversion methods.

$$Upper = Q_3 + 1.5(Q_3 - Q_1)$$
$$Lower = Q_1 - 1.5(Q_3 - Q_1)$$

$$(19)$$

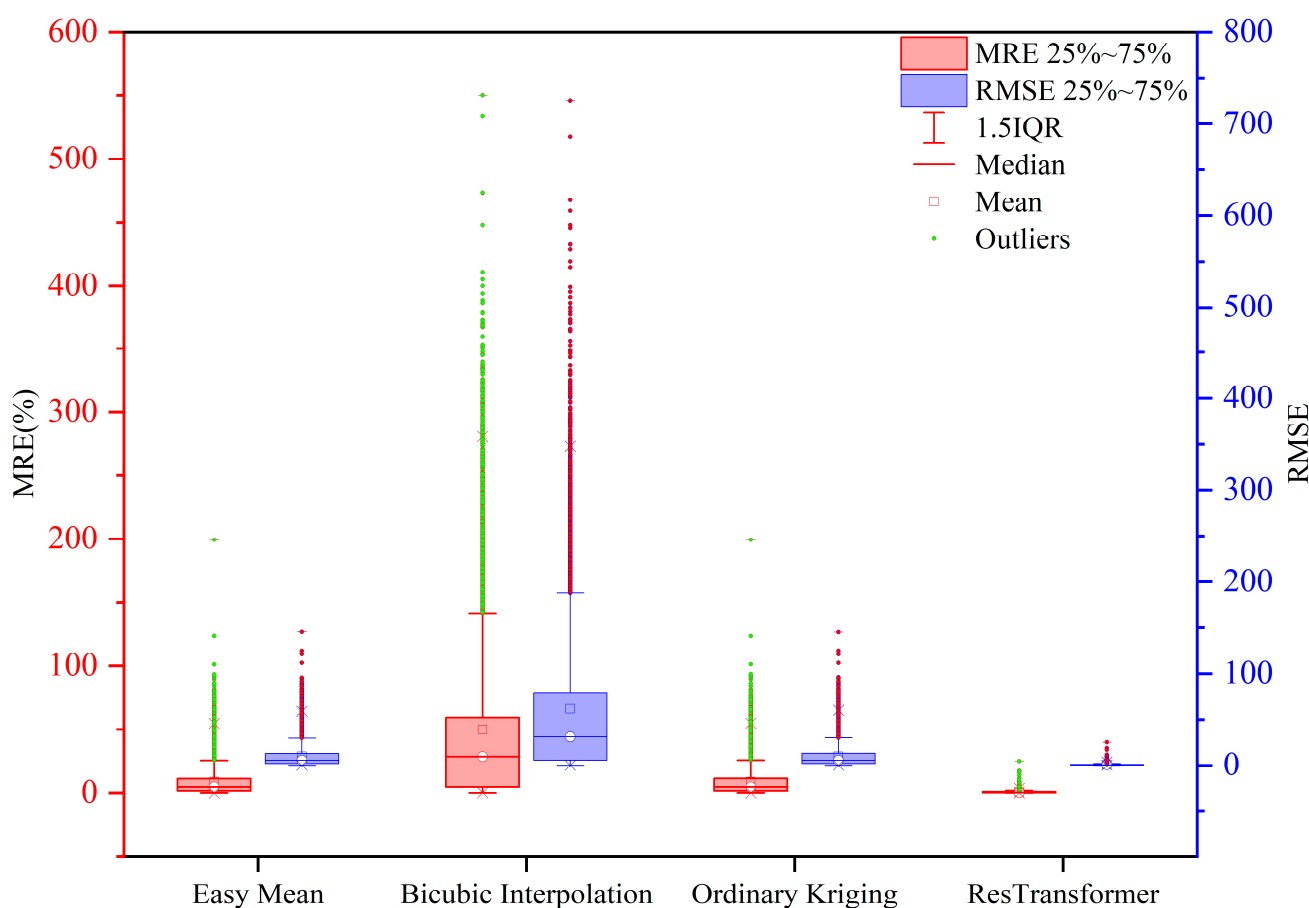

**Figure 11.** Distribution of MRE and RMSE as a result of various scale conversion methods. The solid dots in the graph indicate outliers exceeding 1.5 times the IQR, and the hollow circles indicate the average values.

The results of the evaluation metrics calculated for each scale conversion method are shown in Table 3. The results show that among these four methods, the Simple Average and Ordinary Kriging methods have relatively high average MRE and average RMSE values of 8.56704 and 9.8842, and 8.60315 and 9.9272, respectively. This indicates that the mean error and root mean square error of these two methods are more considerable, i.e., they have lower prediction accuracy. However, both methods' median MRE and median RMSE values are relatively low, indicating that they can yield relatively accurate predictions in some cases. The Cubic Spline Interpolation method is one of the worst performing methods with mean MRE and mean RMSE values of 49.77202 and 61.78559, respectively, which means that it has a very high mean error and root mean square error and is therefore not suitable for accurate prediction.

**Table 3.** Average accuracy of various scale conversion methods on the SCD dataset.

| Method Name | Avg MRE (%) | Avg RMSE | Avg IQR MRE (%) | Avg IQR RMSE | Median MRE (%) | Median RMSE | R |
|---|---|---|---|---|---|---|---|
| Simple Average | 8.56704 | 9.8842 | 6.16947 | 7.43261 | 4.58799 | 5.67452 | 0.85693 |
| Cubic Spline Interpolation | 49.77202 | 61.78559 | 31.22144 | 39.93744 | 28.46038 | 31.0651 | 0.40127 |
| Ordinary Kriging | 8.60315 | 9.9272 | 6.20061 | 7.49048 | 4.62784 | 5.71912 | 0.85527 |
| ResTransformer | 0.6440 | 0.7460 | 0.52335 | 0.62297 | 0.6440 | 0.5490 | 0.99911 |

Moreover, it also has high IQR MRE and RMSE values, indicating that its predictions are less stable. In addition, its median MRE and median RMSE values are also high,

indicating that the method generates significant prediction errors in some cases. In contrast, the mean MRE and RMSE values of the ResTransformer method are 0.6440 and 0.7460, respectively, which means that it has a low mean and mean square root error. At the same time, the ResTransformer model on Avg MRE, Avg IQR MRE, Avg RMSE, and Avg IQR RMSE all decreased by more than 90% compared to the baseline. The median decrease in MRE and RMSE was more significant than 85%, and the correlation coefficient improved by 16.59% compared to the baseline. The ResTransformer model had the best performance in all parameters relative to the baseline method. In addition, the ResTransformer method has the smallest IQR MRE and IQR RMSE values, indicating that the stability of its prediction result accuracy is also good. Overall, ResTransformer is the best method. Simple Average and Ordinary Kriging can provide better prediction results in some cases, while the Cubic Spline Interpolation method performs the worst.

### 4.2. Effect of the Number of Sampling Points and Sample Area on the Accuracy of Scale Conversion Results

In surface reflectance measurement, the accuracy of surface reflectance at the image scale will be changed when different sampling methods are used for the same size sample area and the same sampling methods are used for different size sample areas. The average MREs of the traditional medium- and high-resolution satellite image size samples were calculated under different standard sampling methods, including 30 m, 20 m, 10 m, 8 m, 4 m and 2 m, corresponding to Landsat, Spot, Sentinel and GF series satellites. The standard sampling methods include one-point, two-point, four-point, five-point, nine-point, and sixteen-point sampling methods.

Figure 12 shows the scale conversion results' accuracy with different sampling points and sample areas. The figure shows that the scale conversion results of the Simple Average method and ResTransformer model, in the same size of a sample area, the MRE gradually decreases as the number of sampling points increases. When the sampling method is fixed, the MRE does not strictly increase with the increase of sample area, and the MRE is in a fluctuating rising state. We found that the error of the same scale conversion method in the 4 m sample area is much higher than that in the 20 m sample area under the sampling method with a smaller number of sampling points. Therefore, the number of sampling points is the main factor affecting the accuracy of the scale conversion results in the actual sampling process.

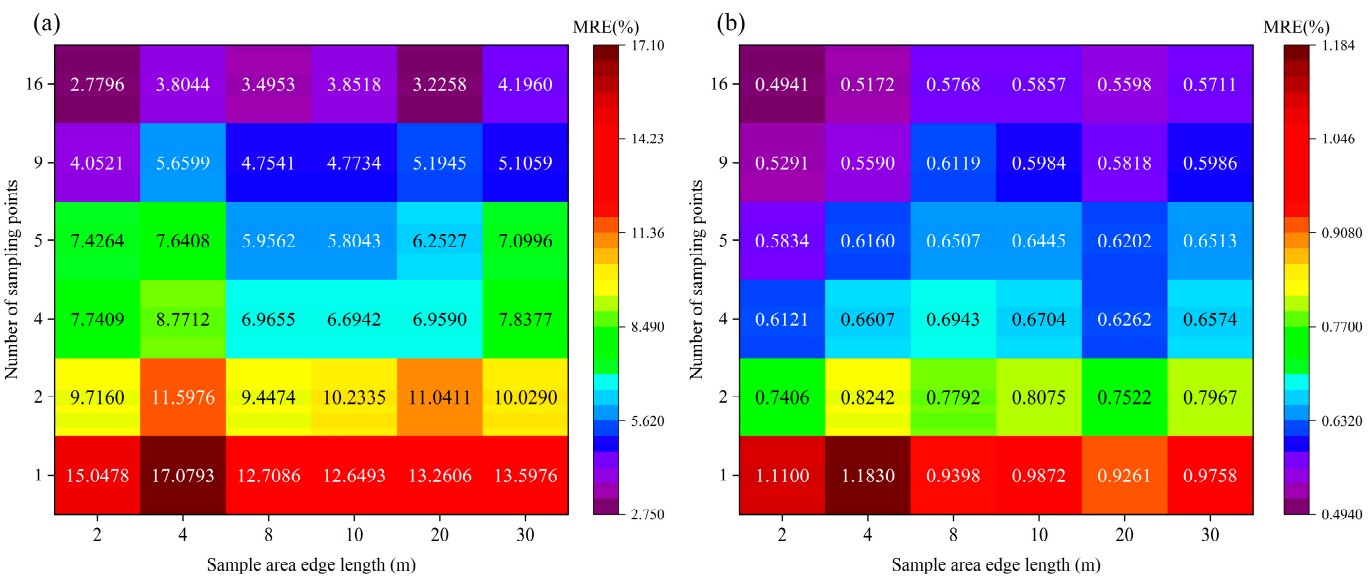

**Figure 12.** Accuracy of scale transformation results with different numbers of sampling points and sample area. (**a**) Simple Average method. (**b**) ResTransformer model.

Tables 4 and 5 quantify the influence of the number of sampling points and sample area size on the accuracy of the scale conversion results by calculating the correlation coefficients between the number of sampling points and the MRE of the scale conversion results under the same size sample area and the correlation coefficients between the sample area size and the MRE of the scale conversion results under the same number of sampling points. The results revealed that the MRE of the scale conversion results showed an apparent negative correlation with the number of sampling points. In contrast, the correlation with the sample area was not significant, especially when the fixed number of sampling points was 2 and 4, and there was no linear correlation between the MRE of the scale conversion results and the sample area size. Therefore, in the actual sampling process, the number of sampling points was the main factor affecting the accuracy of the scale conversion results.

**Table 4.** Correlation between the number of sampling points and the scale conversion result MRE under the same size sample area.

| Scale Conversion Method | Sample Area Edge Length (m) | R | Scale Conversion Method | Sample Area Edge Length (m) | R |
|---|---|---|---|---|---|
| Simple Average | 2 | −0.8618 | ResTransformer | 2 | −0.7048 |
| | 4 | −0.8431 | | 4 | −0.7299 |
| | 8 | −0.8455 | | 8 | −0.7987 |
| | 10 | −0.8211 | | 10 | −0.7461 |
| | 20 | −0.8626 | | 20 | −0.7371 |
| | 30 | −0.8526 | | 30 | −0.7627 |

**Table 5.** The correlation between sample area size and scale conversion results MRE under the same number of sampling points.

| Scale Conversion Method | Number of Sampling Points | R | Scale Conversion Method | Number of Sampling Points | R |
|---|---|---|---|---|---|
| Simple Average | 1 | −0.4412 | ResTransformer | 1 | −0.6118 |
| | 2 | −0.0008 | | 2 | 0.0346 |
| | 4 | −0.1703 | | 4 | 0.0061 |
| | 5 | −0.1515 | | 5 | 0.5472 |
| | 9 | 0.3204 | | 9 | 0.5243 |
| | 16 | 0.5300 | | 16 | 0.5608 |

In the field image-scale geo-spectral acquisition activities, if we need to carry out star–earth synchronous matching calibration of various satellite sensors through the measured image scale geo-spectra, then we need to increase the time cost, reduce the efficiency, and increase the number of sampling points as much as possible to obtain the image scale geo-spectral data with higher accuracy. Field image-scale geo-spectral acquisition also includes automatic instrument acquisition, which usually involves installing automatic geo-spectral acquisition devices at fixed locations. Since it is costly to add automatic acquisition instruments, the number of instruments is fixed. The accuracy of image scale geo-spectral can only be improved by improving scale conversion methods.

During routine field spectral acquisition, too many sampling points will lead to an increase in the chance of the sample area being destroyed by human factors and an increase in the uncertainty caused by human factors. This means that the accuracy of the scale conversion results cannot be improved by increasing the number of sampling points, so it is necessary to achieve the least number of sampling methods while ensuring a certain accuracy. As can be seen from Figure 12, when the number of sampling points is increased from 1 to 4, the ResTransformer scale conversion method results in an average decrease of 35.3% in MRE and a 4-fold increase in workload, and when the number is increased from 1 to 16, the scale conversion results in an average decrease of 45.3% in MRE, but a 16-fold

increase in sampling workload. Similarly, when the number of sampling points is increased from 1 to 4, the MRE of the Simple Average scale conversion method results in an average decrease of 33%, and when the number is increased from 1 to 16, the MRE of the scale conversion results in an average decrease of 74.3%. As the number of samples increases, the rate of MRE decreases, slows down, and the sampling efficiency decreases. Therefore, when the accuracy of four-point sampling meets the required conditions, increasing the number of additional sampling points becomes unnecessary.

### 4.3. Effect of Different Sub-Bedding Surfaces on the Accuracy of Scale Conversion Results

In the process of surface reflectance field collection, the accuracy of the scale conversion results will change due to the different heterogeneity of different feature types. To investigate the influence of different feature types on the scale conversion accuracy, the Simple Average and ResTransformer methods were used to scale convert 34 different feature types in the SCD dataset, and the average MRE values of the scale conversion results of different feature types were calculated.

Figure 13 shows the average accuracy of the scale conversion results for different subsurface samples. The figure shows that the Simple Average method and the ResTransformer model have similar trends in the accuracy of the scale conversion results for different feature types. Meanwhile, the figure shows that the accuracy of the scale conversion results obtained by different scale conversion methods is higher for the feature types with homogeneous ground surfaces and low ground-cover height. The typical features of this type include non-vegetated ground surfaces such as water bodies, asphalt, concrete, bare soil and sand, and intensive vegetation covers such as grass and straw. For the type of surface cover with higher vegetation and dense vegetation, rarely seen bare soil between the monopoly, the accuracy of the scale conversion results of different scale conversion methods is higher compared with the types of sand and water bodies. However, the number of sampling points can be increased to improve sure accuracy, and the typical features of this type include dense vegetation cover such as cotton, sweet potato, okra, peanut and soybean. For monopoly row vegetation, higher crops and sparse, non-uniformly distributed ground types, the accuracy of the scale conversion results is lower. Typical ground types include corn, weeds and vegetation such as red-fruited canary.

To quantitatively describe the influence of spatial heterogeneity on the accuracy of scale conversion results of various scale conversion methods, the coefficients of variation of different samples were calculated by Equation (18), where $std_{DN}$ and $mean_{DN}$ are the standard deviation and mean of $DN$ values of UAV grayscale images in the sample area, respectively. The correlation between the coefficients of variation and the accuracy of scale conversion results was checked to quantitatively describe the influence of spatial heterogeneity on the accuracy of scale conversion results of various scale conversion methods.

$$cv = \frac{std_{DN}}{mean_{DN}} * 100\% \tag{20}$$

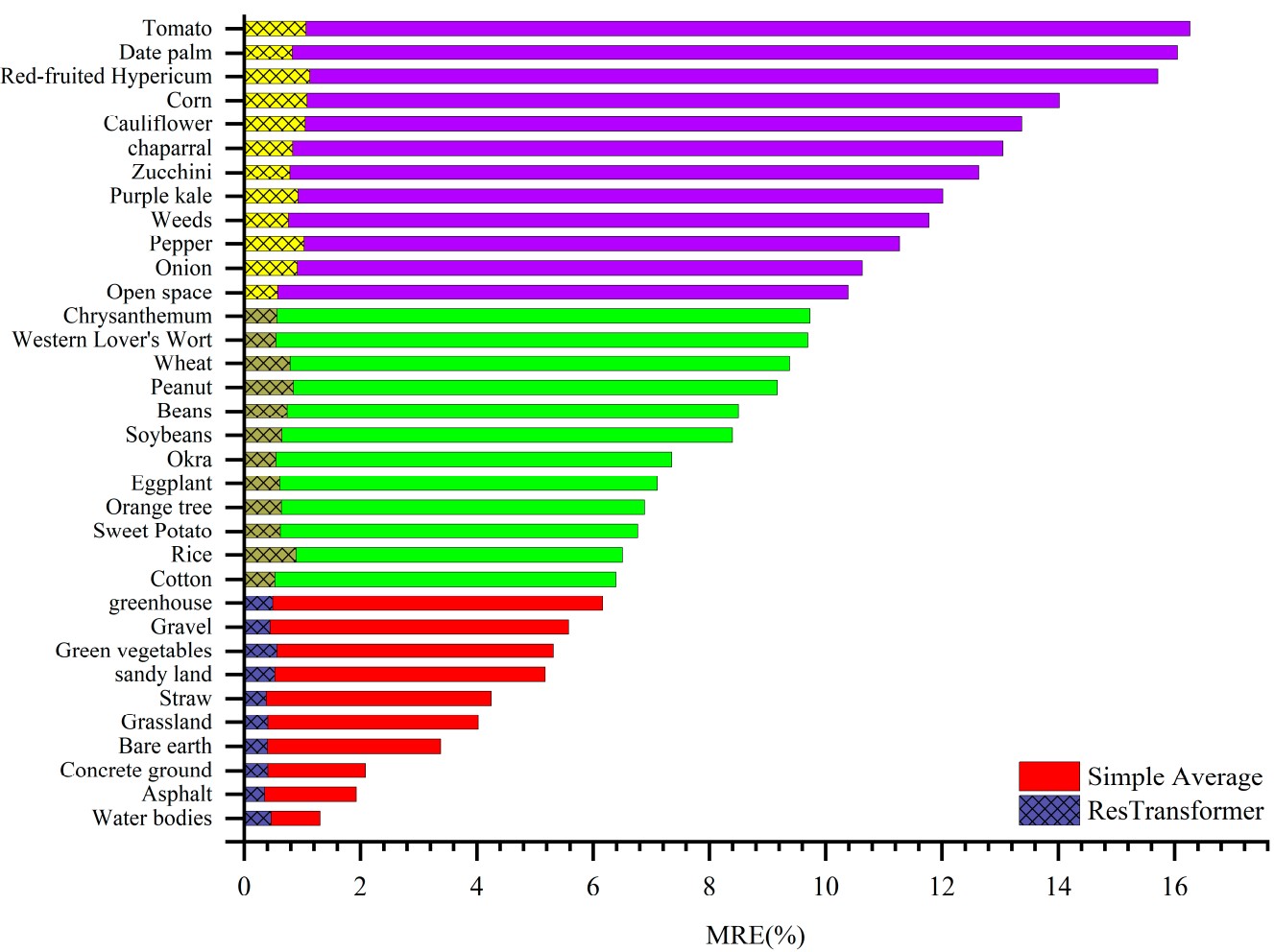

**Figure 13.** Average precision of the conversion results of sample scales for different feature types. The red, green, and blue bars in the figure represent the types of surface coverage with high, medium, and low accuracy for surface reflectance scale conversion, respectively.

Figure 14 shows the correlation between the surface heterogeneity and the accuracy of the scale transformation. The results in the figure show that the linear relationship between the MRE of the Simple Average method and the ResTransformer model results and the coefficient of variation of the sample area is more significant, and the $R^2$ of the linear fit is 0.75, indicating that the accuracy of the scale transformation method results decreases gradually with the increase of the surface heterogeneity. The $R^2$ of the linear fit is 0.75, indicating that the accuracy of the scale conversion method decreases as the surface heterogeneity increases. Meanwhile, the MRE of the Simple Average method changes rapidly with the increase in coefficient of variation of the sample area, and the slope of the fitted straight line is 0.21. The MRE of the ResTransformer model changes slowly with the increase in coefficient of variation of the sample area, proving that the ResTransformer model has higher robustness and accuracy than the Simple Average method. It is proved that the ResTransformer model has higher robustness and universality compared with the Simple Average method and can adapt to scale transformation tasks in more complex sub-bedding scenarios.

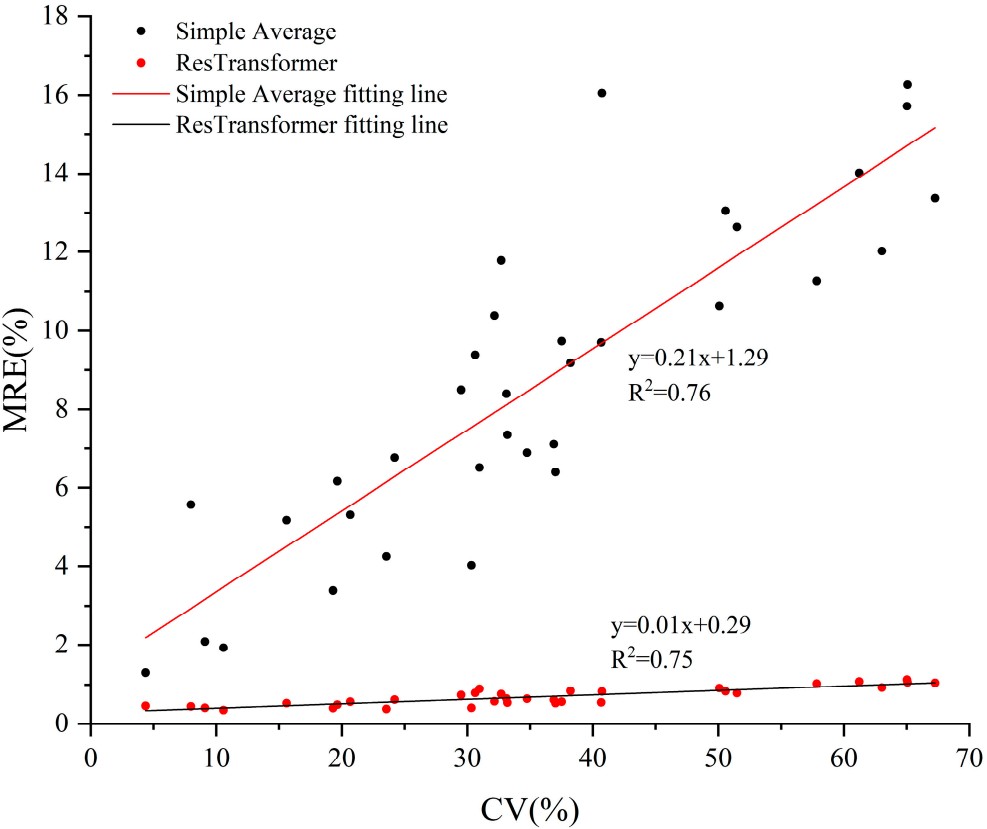

**Figure 14.** Correlation between the coefficient of variation of the sample area and the MRE of the scale conversion results of various scale conversion methods.

## 5. Conclusions

In this paper, we convert the scale conversion problem of surface reflectance to the scale conversion problem of grayscale values of sampling points in UAV images by imputation and prove the correctness of imputation. This is achieved by establishing a functional relationship between the two, and employing the ResTransformer deep learning model to extract and fuse features. The ResTransformer deep learning model is also established to extract, fuse, and adaptively learn the features within the UAV image of the target sample area and the UAV image of the sample points within the sample area, and determine the high-dimensional nonlinear spatial relationship between the sample points and the sample area within the target sample area. This enables realization of the scale conversion of the surface reflectance at the image element scale with high accuracy under the scenarios of arbitrary size, heterogeneous sample area and sampling mode. To verify the accuracy and robustness of ResTransformer, a scale-transformed dataset including 500k samples is built in this paper. Moreover, after being validated through various traditional methods with ResTransformer, the results show that the ResTransformer deep learning model scales the transformation results on the SCD dataset with much better accuracy than the Cubic Spline Interpolation and Ordinary Kriging methods, while compared to the Simple Average method, the baseline is greatly improved. The mean MRE, mean MRSE, and correlation coefficient R with the actual value for ResTransformer on the SCD dataset were 0.6440%, 0.7460, and 0.99911, respectively.

In comparison, the accuracy of the baseline on the SCD dataset was 8.5670%, 9.8842, and 0.8569, respectively. The improvements of ResTransformer accuracy compared to the baseline were 92.48%, 92.45% and 16.59%, respectively. Compared with the baseline method, ResTransformer offers high robustness and is suitable for different sampling methods and sample areas of different sizes. In contrast, the accuracy of the baseline method changes significantly with a decrease in the number of sampling points and an

increase in the sample area. Hence, the method is only suitable in certain applications. In future research, we plan to iteratively update the model by designing and building a large model as well as creating a large sample dataset with big data. Further, we plan to extend the imputation method, as well as the model, from surface reflectance to other surface quantitative remote sensing parameters, such as surface temperature and LAI.

**Author Contributions:** Conceptualization, funding acquisition, writing—review and editing, project administration, H.G.; Conceptualization, methodology, writing—original draft preparation, software, validation, X.Q.; investigation, Y.W.; resources, F.L., Y.Y., Z.L. and W.Z.; data curation, X.S.; visualization, supervision, Q.W.; formal analysis, X.Z. All authors have read and agreed to the published version of the manuscript.

**Funding:** This research was funded by the Strategic Priority Research Program of the Chinese Academy of Sciences (XDA28050401), China's 13th Five-Year Plan Civil Space Pre-Research Project (EOA203010F) and Ecological environment satellite star-ground synchronous authenticity verification experiment (E2C2031501).

**Data Availability Statement:** Not applicable.

**Conflicts of Interest:** The authors declare no conflict of interest.

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
