# Peer review of "A Scale Conversion Model Based on Deep Learning of UAV Images"

_remotesensing, doi:10.3390/rs15092449_

Round 1

Reviewer 1 Report

Contribution/Summary: This study introduces ResTransformer, a deep learning model for scale conversion of surface reflectance using UAV photos. ResTransformer fully extracts and fuses features of UAV images in the sample area and sample points and establishes a high-dimensional nonlinear spatial correlation between sample points and sample area in the target sample area, enabling quick and accurate scale conversion of surface reflectance at the pixel-scale. The authors collected 500k samples to test the model's accuracy and resilience against alternative scale conversion methods. ResTransformer beats other approaches in accuracy, robustness, and universality, making it a potential pixel-scale surface reflectance scale converter.

Comments/Suggestions:
A. Some Advantages:

1. ResTransformer is a promising, highly accurate, and robust method for converting pixel-scale surface reflectance scale using UAV images.

2. The model fully extracts and fuses features of UAV images in the sample area and sample points, establishing a high-dimensional nonlinear spatial correlation between sample points and sample area in the target sample area, enabling quick and accurate scale conversion of surface reflectance at the pixel-scale.

B. What are the limitations and challenges associated with traditional point-to-surface conversion methods in acquiring pixel-scale surface reflectance?

C. Can you explain in more detail how the nonlinear correlation model is established between the UAV image in the sample area and the UAV image in the sampling point, and how it is applied to complete the reduction of the reflectance scale conversion problem?

D. The authors should include a paragraph on how formal methods might be used to verify AI-based solutions, notably data collection and processing. For this purpose, the following references may be included:

  a. https://ieeexplore.ieee.org/document/9842406   b.  https://dl.acm.org/doi/abs/10.1145/3503914

E. Please explain how sample enhancement is achieved in this study, why it is necessary to simulate UAV imaging at different heights, and provide a detailed explanation of the equations used to calculate the side length width of the square sample area and the corresponding sample area.

F. Can you provide more details on how the average DN values of the sample point and the sample area UAV grayscale images are calculated to construct a single SCD sample? Additionally, can you explain how these single samples are then combined to create the dataset used for training and testing the ResTransformer model?

G. Please explain in detail how the mean relative error (MRE), root mean square error (RMSE), and correlation coefficient (R) are calculated in order to compare the results of the ResTransformer model to those of the Simple Average method, Cubic Spline Interpolation, and Kriging Interpolation. Can you also elucidate how the actual value of the SCD data set (DNP) and the outcome of the scale conversion method (DNPP) are utilized in these equations?

H. Please describe the effects of various sampling techniques and sample area sizes on the precision of surface reflectance measurements at the picture scale, as well as how these effects vary across various satellite image resolutions.

I. What is the relationship between the number of sampling points and the accuracy of surface reflectance measurements at the image scale, and how does this affect the workload and efficiency of scale conversion methods? Can the accuracy be improved by increasing the number of sampling points, and at what point does the rate of improvement slow down or become unnecessary?      

Acceptable.

Reviewer 2 Report

A Scale Conversion Model Based on Deep Learning of UAV Images

Few specific facts about this manuscript are as follows:

1. How optimization is achieved in the ResTransformer model ? Please justify it into the manuscript.

2. Generally in the deep residual network, an increase in the network depth increases the training error, leading to an increment in the test error. How ResTransformer model handles this? Please justify it into the manuscript.

3. How is the overfitting and bias problem handled in Transformer while utilizing the proposed  ResTransformer model? Please justify it into the manuscript.

4. For a deeper network, the detection of errors is very difficult for ResNet and learning may be inefficient, how this scenario is handled in the proposed ResTransformer model? Please justify it into the manuscript.

5. A detailed flow chart should be included in the manuscript?

6. Limitations and future research directions should be included.

7. Please include complexity analysis also. 

Minor editing is required, and some sentences should be reframed clearly in the introduction section for better understanding to the readers.

Reviewer 3 Report

In this paper, the authors develop a deep learning model with UAV images to achieve high-accuracy pixel-scale surface reflectance scale conversion for arbitrary size, heterogeneous sample area, arbitrary sampling location and the number of scenes. In general, the work in this paper is well described. However, the following two queries the authors should be addressed.

1. In Eq. (3), the variable k should be explained.

2. In line 225 on page 8, the IFOV is set to 25°. The reviewer wanders to know how to set this parameter. The authors should clearly clarify the method.

Round 2

Reviewer 1 Report

The authors considered my comments and suggestions. Good luck.

Acceptable

Reviewer 2 Report

A Scale Conversion Model Based on Deep Learning of UAV Images

Authors have addressed almost all the issues, the manuscript in the current state may be considered for the publication. 

Some sentences may be revisited for more clarity.